# What gets a dentist hired? Factors influencing employment of recent dentist graduates in Saudi Arabia

Khalid Talal Aboalshamat[1¤a]*, Mohammed Osamah AlHajeeri[2¤b],
Mohammad Abdullah Al Abdullah[2¤b], Khalid Mohammed Alzahrani[2¤b],
Ammar Abdulmohsen Allehyani[2¤b]

**1** Dental Public Health Division, Preventive Dentistry Department, College of Dentistry, Umm Al-Qura University, Makkah, Saudi Arabia, **2** College of Dentistry, Umm Al-Qura University, Makkah, Saudi Arabia

¤a Current address: Preventive Dentistry Department, College of Dentistry, Umm Al-Qura University, Makkah, Saudi Arabia
¤b Current address: College of Dentistry, Umm Al-Qura University, Makkah, Saudi Arabia
* Ktaboalshamat@uqu.edu.sa

## Abstract

In Saudi Arabia, the media have highlighted employment struggles among newly graduated dentists. The aim of this study was to assess the factors that influence the employment of dentists who graduated within the last 5 years. This was a cross-sectional study that assessed 241 dentists who recently graduated from dental studies in Saudi Arabia. The assessment was conducted using a validated self-reported questionnaire. Among the participants, 66.39% were employed (20.75% in the governmental sector, 1.66% in academics, 43.98% in the private sector). Employment was associated with being married ($p = 0.047$), graduated more than a year ago ($p < 0.001$), and who had sales skills ($p = 0.015$) in both government and private sectors. Graduates from private universities were associated with employment in the private sector ($p = 0.004$). Graduates with a higher grade point average (GPA) were associated with employment in government jobs, while lower GPA graduates were associated with private sector employment ($p = 0.006$). Publishing scientific papers increased employment odds by 53.5% ($p = 0.001$, $R^2 = 0.071$). Applying for government or academic jobs improved odds by 13.4% ($p = 0.006$, $R^2 = 0.060$). Postgraduate applications to the Saudi Commission of Health Specialties (SCHS) raised employment chances by 47.5% ($p = 0.002$, $R^2 = 0.061$). Dentists in government or academic jobs were more satisfied with their salaries (79.61%, $p < 0.001$) than were private sector employees (26.41%). Dental employment levels in Saudi Arabia are surprisingly better than expected. However, more efforts are needed to improve the employment percentage and increase salary satisfaction in the private sector.

**Data availability statement:** All relevant data are available within the paper and its Supporting information files.

**Funding:** The author(s) received no specific funding for this work.

**Competing interests:** The authors have declared that no competing interests exist.

**Abbreviations:** GPA, Grade point average; SCHS, Saudi Commission for Health Specialties; SD, Standard deviation; SDLE, Saudi Dental Licensure Examination.

## Introduction

For many years, young students globally have viewed dentistry as a promising career option, primarily due to its financial stability and favorable job prospects [1,2]. However, dentistry as a career seems to encompass a variety of situations in different countries around the world. For example, in the United Kingdom and the United States, there is a high demand for quality professionals [3,4]. The standards, however, might be different in other countries, such as China [5] and Saudi Arabia [6]. In China, there has been a drastic increase in the number of dental schools and undergraduate students, which has resulted in more hiring competition among dentists [5]. There, dentists have difficulty finding an ideal practice position after graduation [5]. The main contributing factors in these difficulties are lack of employment opportunities, lack of work experience, and graduates holding only an undergraduate degree [5]. A similar situation was observed in Saudi Arabia, and media reports have highlighted the struggle to find appropriate employment among newly graduated dentists in the Kingdom [7]. Also similarly, the situation has been attributed to the increased number of dental schools in Saudi Arabia [7]. In fact, according to the latest statistics from the Saudi Ministry of Health [8], there are 25,970 dentists in Saudi Arabia. Of these, 53.74% are Saudi nationals. The workforce is divided between the governmental (35.78%) and private sectors (64.21%). Among dentists in the governmental sector, 90.41% are Saudi, while only 33.30% of dentists in the private sector are Saudi nationals, as shown in Table 1.

It should be mentioned that there is no clear data regarding unemployed dentists in Saudi Arabia; however, the media have reported that the number was approximately 7,000 dentists without jobs in Saudi Arabia in 2023 [9].

In Saudi Arabia, there are 27 dental colleges, with 17 that are governmental and 10 that are private [8]. In the governmental dental colleges, there were 5,562 students and 1,151 graduates in just 2023 [8]. There is no available data for the numbers of students and graduates in private dental schools in Saudi Arabia; however, the estimation is likely to be higher than the governmental numbers.

New graduates in dentistry have different available future opportunities, including continuing on to a postgraduate degree, working in government, or working in the private sector [6]. Two studies in Nigeria have shown that most undergraduate dental students prefer to pursue further education in a specialty program [10,11]. In contrast, a study conducted in the United States found that 58.9% of participants intended to start working in private practice right after graduating from dental school [12]. In Saudi Arabia, half of the dental graduates pursue postgraduate training, with many choosing to enroll in training programs immediately following graduation [13]. In Saudi Arabia, governmental dental positions are more desirable for recent graduates because they provide better salaries and less competition than the private dental clinic market [6]. However, the increasing number of dentists has intensified competition for the limited numbers of government positions and specialty postgraduate program seats [6]. For private practice, many students in Dammam, Saudi Arabia, face challenges in securing employment after graduation due to a highly competitive

**Table 1. Distribution of dentists in Saudi Arabia according to Ministry of Health (2023).**

| Sector | Saudi *n* (%) | Non-Saudi *n* (%) | Total *n* (%) |
|---|---|---|---|
| Government | 8,403 (90.41%) | 891 (9.59%) | 9,294 (100%) |
| Private | 5,554 (33.30%) | 11,122 (66.67%) | 16,676 (100%) |
| Total | 13,957 | 12,013 | 25,970 (100%) |

job market [6]. They also report difficulties encountered in establishing a private practice [6]. This is supported by a study from Riyadh, Saudi Arabia, that found 62% of respondents believed that setting up a private dental practice would require a considerable amount of time and effort [14].

There have been some distinguished efforts to increase the Saudi employment status conducted by higher organizational entities, such as the announcement by the Saudi Human Resources and Social Development of an initiative to raise the percentage of Saudi dentists working in the private sector to 35%, starting on March 10, 2024 (Ministry of Human Resources and Social Development, 2025). This initiative aimed to provide more job opportunities in the private sector for Saudi dentists in different regions of the Kingdom [15].

Studies in the literature highlight some factors that affect dentists' employment chances. A recent study in Jeddah, Saudi Arabia, indicated that dentists were able to find a job immediately after graduation [16]. For example, a majority of graduates (75%) obtained employment within the first year after graduation, and most were within the first 6 months [16]. It should be noted that the authors in that study stated that their questionnaire did not investigate the underlying reasons for this trend [16]. In a study among dental students and interns in Saudi Arabia, the factors of being female, having low academic grades, and low education levels of the parents were found to impose more challenges in finding a dental job [6]. However, in Jeddah, Saudi Arabia, there was no effect of gender, grade point average (GPA), or Saudi Dental Licensure Examination (SDLE) scores on employment opportunities among graduated dentists [16]. The difference between the two studies might be due to differences in the investigated cities or because one study was conducted among students while the other included graduated dentists.

## Aim

To the best of our knowledge, no study has assessed factors that influence the employment of recently graduated dentists across different regions in Saudi Arabia, nor have any investigated other factors related to personal and soft skills useful for getting a job. Thus, the aim of this study was to assess the factors that influenced the employment of dentists in Saudi Arabia who graduated within the last five years.

## Materials and methods

Participants in this study were recruited using convenience sampling for this cross-sectional study from 11 November 2024 to 1 December 2024. Because contact information and relevant data for the target population were unavailable, constructing a sampling frame was impractical, therefore convenience sampling was used. The inclusion criteria were dentists who graduated from a Saudi dental school within the last 5 years (between 2020 and 2024), with at least 4 months since graduation. The requirement of a minimum of 4 months post-graduation was applied because, at the time of recruitment, the most recent graduates at year of 2024 in all universities in Saudi Arabia, had just completed their internship year and were eligible to apply for jobs. This ensured that only fully graduated dentists were included and prevented potential confusion among individuals who had not yet officially graduated. The participant had to be a resident of Saudi Arabia, whether employed or not. The exclusion criteria were dental students, interns, and those who did not sign the informed consent form supplied prior to the start of the study.

The required data were collected with a questionnaire that was sent through social media applications, including WhatsApp, Instagram, Twitter (X), Snapchat, and Telegram. Recruitment primarily involved identifying and joining groups that included dentists in Saudi Arabia, either through keyword searches or referrals from colleagues and friends. Study invitations were then shared within these groups, both through open posts and direct messages to individual members. In addition, potential participants were identified on platforms such as Twitter/X and LinkedIn by searching relevant keywords and reviewing user profiles or connections, after which invitations were sent privately to dentists who met the inclusion criteria. No paid or sponsored advertising was used.

The questionnaire used in this study comprised 46 questions that were developed after reviewing the body of literature, and some of the questions were adapted from previous research [5,6,16,17]. However, the majority of questions were developed for this study by five dental professionals (one professor of dental public health and four recent dental interns) after multiple discussion panels. These items included number of publications/volunteering and job applications (government, private, Saudi Commission for Health Specialties, and non-dental); willingness/constraint items (relocation, rural work, extra hours with/without pay, salary delay, limited tools, working as a dental assistant, unlicensed procedures); social-media activity/followers; soft-skills/entrepreneurship; and income model (fixed, percentage, or both). Some sensitive questions were added based on real field events, despite thier ethical sensitivity, such as "If your clinical manager asked you to do a procedure you were not licensed to do (e.g., orthodontic treatment), would you do it?". This ethically sensitive item was reframed to measure perceived pressure rather than willingness, with anonymous responses interpreted as exploratory. The questions were corrected and adjusted multiple times for clarity and then organized into appropriate sections of the questionnaire. The questionnaire was then sent in a pilot round to 17 participants (dental professionals) to validate the content, understanding, logic, syntax, flow, spelling, and grammar. Feedback led to simplify items' wording and instructions, expand and standardize response options to avoid confusion and overlapping. Some items were removed based on feedback on the length of the original questionnaire, to reduce response burden. Other items were reordered to improve logic and flow. This step was mandatory to ensure the face validity of the questionnaire. Editing was conducted twice before the final version of the questionnaire was developed.

These questions are presented in four sections. Section one includes 13 questions collecting demographic data, including gender, age, nationality, region, city of residence, marital status, university type, graduation year, GPA, source of income, financial responsibility, family members who were dentists, and employment status. Section two presented 18 questions investigating academic and career factors that might affect employment potential. This section asked about the number of scientific publications completed, volunteer work done, job applications submitted in the government and private sectors, postgraduate degree, and job applications submitted in fields not related to dentistry. In addition, this section asked about the willingness of participants to accept specific sets of circumstances when working in the dental field and about any preferences for field of work. There were two questions about the minimum acceptable salary and what is considered a fair salary to the participants in the dental field. In this section, responses were either a numerical reply written in or a yes/no answer. Section three included 11 questions to assess the soft skills that affect employment. This section asked about social media activity and personal soft skills, such as accounting, social media influencing, content creation, photography, communications, business skills, sales, marketing, and entrepreneurial skills. The responses for this section were yes or no. The final section was answered only by employed respondents and encompassed four questions that included information regarding current job section, mode of employment, type of salary, and salary satisfaction.

The participants took approximately 10–15 minutes to complete the survey. The collected data were analyzed with SPSS software (IBM Corp., Armonk, NY, USA). A $p$-value of 0.05 was used as the significance level. Mean, standard deviation, count, and percentages were generated as descriptive statistics, and logistic regression was used to analyze the data. The study was approved by the institutional review board of Umm Al-Qura University, Makkah, Saudi Arabia, with the number HAPO-02-K-012-2024-10-2314. Informed consent was obtained electronically: participants were required to click the "approve" button on the digital consent form before gaining access to the questionnaire.

Individuals who did not indicate their agreement were regarded as having declined consent and were not permitted to proceed.

## Results

### Demographic data

Data were collected from 241 participants living in 26 cities in Saudi Arabia, including Abha, Alahsa, Dammam, Aljouf, Alkharj, Alkhobar, Al-Madinah al-Munawwarah, Bahrah, Burydah, Dhahran, Hail, Hofuf, Jamoom, Jazan, Jeddah, Jubail, Khamis Mushait, Majmmah, Makkah, Najran, Qassim, Qateef, Riyadh, Saihat, Unayzah, and Zulfi. The mean (m) age of participants was 26.97 with standard deviation (SD) of 1.95. Out of the total, 160 (66.4%) participants were currently employed, including those working at their first job in dentistry and those with multiple jobs. Meanwhile, 81 (33.6%) were unemployed, including those who have never had a job and those who were previously employed but are not currently working. Participant demographic data are shown in Table 2.

### Descriptive statistics of academic circumstances, career factors, soft skills, and social media interactions that might influence employment

The participants reported many variables that are considered to be academic or career factors that might affect their employment potential. These factors are displayed in Tables 3 and 4. The mean of the reported minimal salary participants were ready to work with as a dentist in the private sector was 8,686.75 Saudi riyals (SD = 3,941.58). In addition, the mean salary that participants considered fair for a recently graduated dentist in the private sector was 10,961.03 Saudi riyals (SD = 9,374.94). Participants also exhibited variability in soft skills, as shown in Table 5. Table 6 presents the differences among participants in social media use.

### Prediction of employment potential by categorical variables

The categorical demographic variables of academic factors, career factors, soft skills, and social media activities were investigated with bivariable analysis using chi-square and Fisher's exact test. The test was conducted to predict participant employment status in governmental/academic sector ($n = 54$, 22.41%), private sector ($n = 106$, 43.98%), or not currently employed ($n = 81$, 33.61%). Only a few variables were found to have statistically significant relationships, including marital status, year of completing internship, graduation university, GPA at graduation, and sales skills, as shown in Table 7. Married individuals ($p = 0.047$), those with earlier internship completion years ($p < 0.001$), and individuals with sales skills ($p = 0.015$) were associated with employment. Those who graduated from private universities were associated with employment in private jobs ($p = 0.004$). Dentists who had a higher GPA were associated with employment in the government sector, whereas those with a lower GPA were associated with employment in the private sector ($p = 0.006$).

The continuous variables, which included published papers, volunteer work, applications to governmental jobs, applications to private jobs, applications to SCHS postgraduate positions, applications to non-dentistry jobs, fair salary consideration, and expected salary, were examined using simple logistic regression to predict if the participant was currently employed ($n = 160$, 66.39%) or not ($n = 81$, 33.61%). Only published papers, applications to governmental jobs, and applications to SCHS postgraduate positions were found to be statistically significant, as follows. Publishing more scientific papers ($p = 0.001$, Exp(B) = 1.535) raised employment odds by 53.5%, accounting for 7.1% of the variance (Nagelkerke $R^2 = 0.071$). Submitting applications for government or academic jobs ($p = 0.006$, Exp(B) = 1.134) increased employment chances by 13.4%, explaining 6% of the variance (Nagelkerke $R^2 = 0.060$). Applications for postgraduate positions at SCHS ($p = 0.002$, Exp(B) = 1.475) raised employment probability by 47.5%, accounting for 6.1% of the variance (Nagelkerke $R^2 = 0.061$).

## Descriptive statistics of employed participants

Among the participants, there were 160 (66.39%) who were currently employed. They were asked several questions regarding their employment, as shown in Table 8. According to a chi-square test, dentists who worked in governmental/academic jobs were statistically more satisfied (79.61%, $p<0.001$) with their salaries than participants working in the private sectors who were satisfied with their salaries (26.41%).

**Table 2. Participant demographic data (N = 241).**

| Variable | Category | *n* | % |
|---|---|---|---|
| Gender | Male | 107 | 44.40 |
| | Female | 134 | 55.60 |
| Nationality | Saudi | 237 | 98.34 |
| | Non-Saudi | 4 | 1.66 |
| Region of Saudi Arabia | Western | 114 | 47.30 |
| | Central | 65 | 26.97 |
| | Eastern | 29 | 12.03 |
| | Southern | 23 | 9.54 |
| | Northern | 10 | 4.15 |
| Marital status | Married | 41 | 17.01 |
| | Non-married | 200 | 82.99 |
| Type of university graduated from | Governmental | 201 | 83.40 |
| | Private | 40 | 16.60 |
| When did you finish your internship year? | 2020 | 37 | 15.35 |
| | 2021 | 45 | 18.67 |
| | 2022 | 27 | 11.20 |
| | 2023 | 76 | 31.54 |
| | 2024 | 56 | 23.24 |
| What was your graduation GPA? | A: Excellent | 113 | 46.89 |
| | B: Very good | 100 | 41.49 |
| | C: Good | 26 | 10.79 |
| | D: Pass | 2 | 0.83 |
| Do you have another source/s of income not related to dentistry? | Yes | 36 | 14.94 |
| | No | 205 | 85.06 |
| Are you partially or fully financially responsible for any family member (spouse, children, parent, etc.)? | Yes | 73 | 30.29 |
| | No | 168 | 69.71 |
| Do you have a senior relative who was a dentist? | Yes | 31 | 12.86 |
| | No | 210 | 87.14 |
| Please describe your career journey | I've had a (single) job, and I am currently employed in that one. | 109 | 45.23 |
| | I've had (multiple) jobs, and I am currently employed in the last one. | 51 | 21.16 |
| | I've had a job (or multiple jobs) in the past, but I am currently unemployed. | 21 | 8.71 |
| | I have not had a job before. | 60 | 24.90 |
| Your current job sector (mainly) | Governmental | 50 | 20.75 |
| | Academic (demonstrator or lecturer) | 4 | 1.66 |
| | Private | 106 | 43.98 |
| | Non-employed | 81 | 33.61 |

**Table 3. Academic and career factors that might affect employment, Part I (N = 241).**

| Variable | Median | Minimum | Maximum | IQR |
|---|---|---|---|---|
| How many scientific papers have you published (please write a number)? | 1 | 0 | 6 | 1 |
| How many volunteer situations have you participated in (please write a number)? | 6 | 0 | 50 | 7 |
| How many times have you applied for a dentist job in the governmental/academic sector (put 0 if you have not applied)? | 2 | 0 | 34 | 4.5 |
| How many times have you applied for a dentist job in the private sector (put 0 if you have not applied)? | 8 | 0 | 160 | 19 |
| How many times have you applied for a postgraduate position in the Saudi Commission of Health Specialties (SCHS)? | 1 | 0 | 5 | 1 |
| How many times have you applied for jobs other than dentistry? | 0 | 0 | 50 | 1 |

IQR: Interquartile range.

**Table 4. Academic and career factors that might affect employment, Part II (N = 241).**

| Variable | Yes *n* (%) |
|---|---|
| Were you willing to work extra hours per day with extra salary? | 195 (80.91) |
| Were you willing to move to another city for a dental job? | 186 (77.18) |
| Were you willing to work in a rural area (village)? | 151 (62.66) |
| Were you ready to work extra hours per day without extra salary? | 58 (24.07) |
| Were you willing to work without the needed tools? | 57 (23.65) |
| Were you willing to work as a dental assistant? | 44 (18.26) |
| Were you comfortable with a delay in salary of 2 months? | 36 (14.94) |
| If your clinical manager asked you to perform a procedure you don't know how to do, would you do it? | 20 (8.30) |
| If your clinical manager asked you to do a procedure you were not licensed to do (e.g., orthodontic treatment), would you do it? | 18 (7.47) |

**Table 5. Participant soft skills that might have a relationship to employment.**

| Variable | Yes *n* (%) |
|---|---|
| Communication skills | 207 (85.89) |
| Photography | 153 (63.49) |
| Business skills | 109 (45.23) |
| Marketing skills | 107 (44.40) |
| Sales skills | 88 (36.51) |
| Content creator | 80 (33.20) |
| Accounting | 75 (31.12) |
| Entrepreneurial skills | 73 (30.29) |
| Social media influencer | 66 (27.39) |

## Discussion

This study interprets employment patterns among recent Saudi dental graduates rather than restating all estimates. The overall picture is that employment was associated with time since internship completion and selected skills (notably sales), with sector differences in salary satisfaction, while several commonly presumed factors (e.g., gender, social media activity) showed no association. These observations are intended to guide hypotheses and further direction of investigations rather than to prescribe policy. In this convenience sample, all results reflect bivariate associations and should be read as preliminary. They are not causal and not population-level estimates.

**Table 6. Participant social media activities and number of followers (N = 241).**

| Category | Instagram | LinkedIn | Twitter (X) | Snapchat | TikTok |
|---|---|---|---|---|---|
| **Activity:** | *n* (%) | *n* (%) | *n* (%) | *n* (%) | *n* (%) |
| I do not have an account | 17 (7.05) | 59 (24.48) | 42 (17.43) | 44 (18.26) | 95 (39.42) |
| No | 80 (33.20) | 97 (40.25) | 119 (49.38) | 125 (51.87) | 97 (40.25) |
| Yes | 144 (59.75) | 85 (35.27) | 80 (33.20) | 72 (29.88) | 49 (20.33) |
| Number of followers: | | | | | |
| I do not have an account | 17 (7.05) | 59 (24.48) | 42 (17.43) | 44 (18.26) | 95 (39.42) |
| Less than 1,000 | 192 (79.67) | 173 (71.78) | 175 (72.61) | 183 (75.93) | 130 (53.94) |
| 1,000–5,000 | 28 (11.62) | 7 (2.90) | 22 (9.13) | 12 (4.98) | 13 (5.39) |
| More than 5,000 | 4 (1.66) | 2 (0.83) | 2 (0.83) | 2 (0.83) | 3 (1.24) |

**Table 7. Prediction of employment status by categorical variables.**

| Variable | Category | Employed in government or academic sector *n* (%) | Employed in private sector *n* (%) | Not employed *n* (%) | *p*-value |
|---|---|---|---|---|---|
| Marital status | Married | 11 (26.83) | 23 (56.10) | 7 (17.07) | 0.047 |
| | Non-married | 43 (21.50) | 83 (41.50) | 74 (37.00) | |
| University of graduation | Governmental | 50 (24.88) | 79 (39.30) | 72 (35.82) | 0.004 |
| | Private | 4 (10.00) | 27 (67.50) | 9 (22.50) | |
| Grade point average (GPA) | A: Excellent | 34 (30.09) | 38 (33.63) | 41 (36.28) | 0.006* |
| | B: Very good | 19 (19.00) | 49 (49.00) | 32 (32.00) | |
| | C: Good | 1 (3.85) | 17 (65.38) | 8 (30.77) | |
| | D: Pass | 0 (0.00) | 2 (100.00) | 0 (0.00) | |
| Year of finishing internship | 2020 | 12 (32.43) | 22 (59.46) | 3 (8.11) | <0.001 |
| | 2021 | 10 (22.22) | 27 (60.00) | 8 (17.78) | |
| | 2022 | 5 (18.52) | 13 (48.15) | 9 (33.33) | |
| | 2023 | 19 (25.00) | 33 (43.42) | 24 (31.58) | |
| | 2024 | 8 (14.29) | 11 (19.64) | 37 (66.07) | |
| Sales skills | Yes | 18 (20.45) | 49 (55.68) | 21 (23.86) | 0.015 |
| | No | 36 (23.53) | 57 (37.25) | 60 (39.22) | |

*Fisher's exact test: Prediction of employment status by categorical variables.

Our data indicated that 66.39% of dentists who graduated from 2020 to 2024 were employed. In contrast, a recent article found that 56% of dentists who had graduated from Jeddah, Saudi Arabia, from 2019 to 2021 were employed [16]. Our results seem higher than the previous study [16], despite the fact that our data included those who graduated most recently in 2024, which is the group with the highest unemployment rate in our results (11.88%). This can be explained by the following. First, our sample (N = 241) was larger than the previous study's sample (N = 100), and we took our sample from different cities across Saudi Arabia, in comparison to graduates from a single institution in the previous study [16]. Second, there are serious initiatives underway to boost the Saudi employment status, conducted by Saudi Human Resources and Social Development and aimed at improving the percentage of Saudi dentists working in the private sector up to 35%. These initiatives began on March 10, 2024 [15]. This might explain the rise in the percentage of employment in our study. It should be noted that the previous study indicated that 42% found jobs within the first year after graduation [16]. Such information was not investigated in our study. Nevertheless, our study highlighted the career journey as another dimension that included being employed in the same job since graduation, having transferred to multiple jobs, working in a

**Table 8. The experience of employed dentists (N = 160).**

| Variable | Answer | n | % |
|---|---|---|---|
| What is your employment status (mainly)? | Full-time dentist | 140 | 87.50 |
| | Part-time dentist | 13 | 8.13 |
| | Dental assistant | 3 | 1.88 |
| | Full-time, but not as a dentist | 2 | 1.25 |
| | Part-time, but not as a dentist | 2 | 1.25 |
| | Owner of a business, but not a dental clinic | 0 | 0.00 |
| | Dental clinic owner | 0 | 0.00 |
| Your current job sector (mainly) | Governmental | 50 | 31.25 |
| | Private | 106 | 66.25 |
| | Academic (demonstrator or lecturer) | 4 | 2.50 |
| If you are working in the dental field, what is the type of income/salary? | Fixed salary | 87 | 54.38 |
| | Percentage | 10 | 6.25 |
| | Both | 59 | 36.88 |
| | I do not work in dental field | 4 | 2.50 |
| Are you satisfied with your salary/income? | Yes | 71 | 44.38 |
| | No | 89 | 55.63 |

non-dental field, and becoming a business owner. The data in our study provide a more insightful overview of the current employment status for dentists in Saudi Arabia.

It is important to indicate that the percentage of employment in our study was higher than what the research team expected. According to a recent article in a Saudi Arabian newspaper, 7,000 Saudi dentists were without jobs in 2023 [9]. Also, social media is full of posts and news regarding unemployment in the dental field in Saudi Arabia, and many view it to be an oversaturated job market. Our data revealed that this view is not realistic and is more pessimistic than reality, as more than 66.39% of recent graduates are employed. This may reflect the efforts of the governmental bodies in Saudi Arabia and entities aiming to improve the employment rate of highly skilled healthcare providers as part of Saudi Vision 2030, with the ultimate goal of improving quality of care.

Our study did not have any dental clinic owners among the participants. This may be explained by the results of a previous study reporting that dentists face difficulties in setting up a private practice [6]. In Riyadh, Saudi Arabia, 62% of respondents believed that starting a private dental clinic would require an excessive amount of time and effort [14]. This can be another direction for stakeholders to study, examining the obstacles and strains involved in setting up a dental clinic. This step can improve the employment rate in the private sector, as each new dental clinic established means more job opportunities for dental graduates.

Only a few variables were found to be predictors of dentists being employed in Saudi Arabia. Married dentists were statistically associated with those who are employed. It is unknown whether marriage is a factor that helps a dentist obtain a job or if the opposite is true. In Romania, individuals were not afraid to get married after gaining employment [18]. Also, in South Africa, employed individuals were more likely to be married [19]. However, it might be possible that dentists who are married have become more responsible and have more familial obligations that made them eager to pursue a job. This association should be interpreted cautiously, as reverse causation or other cultural confounding could explain this relationship. Therefore, the results indicate correlation within this sample and cannot imply causation.

Our study found that there was no significant difference in employment between males and females. This is similar to the previous study [16], but contradictory to another study [6]. The contradiction can be justified by the Fita et al. study being conducted among dental students and interns, but not already employed dentists [6]. In other words, our study and

the Farag et al. study may be more valid results that represent reality. It is commonly believed among the public on social media that females have more chances than males to find employment. However, our results refute this idea. It should be noted that comparisons with prior work, especially Farag et al. [16], should be interpreted cautiously because of differences in design, sampling frame, statistical tests and years of recruitment.

Sales skills was the only soft skill that was significantly related to being employed. This is interesting and supported by another study in the marketing arena that found teaching a sales intervention program helped students find a better job fit [20]. It seems that sales skills can help dentists sell themselves as a product to be hired in private dental sectors. However, it is important to note that our assessment was self-reported, and it is not known how participants rated themselves as having sales skills or not. Further assessment methods and instruments can be used to measure these variables objectively in future research.

Surprisingly, other soft skills that were believed to be important for gaining employment, such as communication skills, photography, business skills, marketing skills, and accounting, were not found to be significant factors for dental employment. More shocking results were that social media activities and number of followers were not found to be significantly associated with being employed. Related studies found social media activity was useful for employed dentists to attract new patients [21], which is a different outcome (patient acquisition among already-employed dentists) rather than initial employability. That finding is supported by another study in Saudi Arabia that indicated social media was being used in personal, professional, and business aspects among currently employed dentists [22]. It seems that social media activity is useful for dentists to gain new patients after employment but not associated with becoming employed in this sample. This could cast more light on the proper role of social media and underscore the limits of its overestimated influence on dentists' careers and futures.

The number of published papers, applications to governmental/academic jobs, and applications to SCHS for postgraduate positions were found to be significant factors for finding employment. In fact, these factors can indicate which endeavors of the applicant as a person increase the likelihood of becoming employed. For example, publishing scientific papers is not part of most undergraduate programs in Saudi Arabia. So, this can be considered as an additional hard milestone to reach. This idea is supported by a study done in the United States among urology residents, which found that spending more time in research during residency was associated with increasing productivity during and after residency [23]. In contrast, the number of applications to private dental jobs and volunteer work were not significant factors. Our data cannot explain this, and more in-depth research is needed to identify the reason for this contradiction.

Our results indicated that a higher GPA meant the dentist was associated with those who are employed in the governmental sector, while dentists with lower GPAs were associated with those who were employed in the private sector. In contrast, Farag et al. [16] did not identify any impact of GPA on the chances of employment. The reason for this might be due to the type of analysis conducted in our study. In fact, when we categorized the dentists in our study to employed and non-employed, we did not find a significant difference. However, when we categorized the dentists into governmental employment, private employment, and non-employed, the result was statistically significant.

Regardless, efforts must be directed to improving the salaries of dentists in the private sector as a continuation of the endeavors to enhance the Saudi dental workforce. In fact, our study showed that 55.63% were not satisfied with their salaries, a percentage that is similar to and supportive of the results of a previous study (46%) [16]. Our results showed a mean minimum salary participants felt they could work with was 8,686.75 Saudi riyals in the private sector, while the mean of a fair salary for a recently graduated dentist in the private sector was 10,961.03 Saudi riyals. The recent obligation of the Ministry of Human Resources and Social Development forced private clinic owners to provide minimum salaries for general dentists of 7,000 Saudi riyals by April 2022 [24]. In fact, another recent announcement raised minimum salaries for general dentists to 9,000 Saudi riyals as of January 2025 and increased the Saudi localization percentage of dentistry in private sectors to 45% [25]. This shows the advancement of Saudi authorities to tackle this issue with high-level decisions. Also, this is important to take into consideration for future amendments aimed at dental workforce

retention in the private sector in light of the governmental direction toward privatization of the health sector, in general, in Saudi Arabia.

The strength of this study comes from obtaining data from several cities across Saudi Arabia, including recent graduates within the last five years. Being one of few studies in Saudi Arabia to identify factors that may influence hiring decisions is a further strength. Nevertheless, the results cannot be generalized to all dentists in Saudi Arabia as the sample technique was not based on probability sampling, and there is a relatively small sample size. Furthermore, the study was based on a self-reported questionnaire, especially those related to soft skills, job applications, and social media activity. Also, as categorical variables were assessed using bivariate tests, multivariable analyses using larger sample size would be required to assess whether the associations persist after adjustment.

Future studies can investigate dental clinic owners or human resources departments in the governmental and private dental sectors to investigate the most important factors leading to dentists becoming employed. Also, it might be of interest to investigate the reasons for hiring private dental college graduates more in the private dental sector as a future research direction. Also, conducting a national study using a probability-based random sample would inform more actionable policy decisions, as our recommendations based on our data are exploratory and should be treated as considerations for pilot testing and further study, not prescriptive policy.

## Conclusion

Surprisingly, two-thirds of recently graduated dentists in Saudi Arabia were employed, primarily in the private sector, while one-third were unemployed. Employment was associated with those who were married, had graduated earlier, had sales skills, achieved higher GPAs, published more scientific papers, and had submitted more applications to government and postgraduate positions. More studies are needed to yield more generalizable results. Stakeholders might take future initiatives to improve salaries in the dental private sector in Saudi Arabia to improve retention and employment.

### Consent for publication

The study was approved by the institutional review board of Umm Al-Qura University, Makkah, Saudi Arabia, with the number HAPO-02-K-012-2024-10-2314.

### Supporting information

**S1 File. Study raw data.**
(PDF)

**S2 File. The study questionnaire.**
(PDF)

### Author contributions

**Conceptualization:** Khalid Talal Aboalshamat, Mohammed Osamah AlHajeeri, Mohammad Abdullah Al Abdullah, Khalid Mohammed Alzahrani, Ammar Abdulmohsen Allehyani.

**Data curation:** Mohammad Abdullah Al Abdullah, Khalid Mohammed Alzahrani, Ammar Abdulmohsen Allehyani.

**Formal analysis:** Khalid Talal Aboalshamat.

**Investigation:** Khalid Talal Aboalshamat, Mohammed Osamah AlHajeeri, Mohammad Abdullah Al Abdullah, Khalid Mohammed Alzahrani, Ammar Abdulmohsen Allehyani.

**Methodology:** Khalid Talal Aboalshamat, Mohammed Osamah AlHajeeri, Mohammad Abdullah Al Abdullah, Khalid Mohammed Alzahrani, Ammar Abdulmohsen Allehyani.

**Project administration:** Khalid Talal Aboalshamat, Mohammed Osamah AlHajeeri, Mohammad Abdullah Al Abdullah, Khalid Mohammed Alzahrani, Ammar Abdulmohsen Allehyani.

**Resources:** Khalid Talal Aboalshamat.

**Software:** Khalid Talal Aboalshamat.

**Supervision:** Khalid Talal Aboalshamat.

**Writing – original draft:** Khalid Talal Aboalshamat, Mohammed Osamah AlHajeeri, Mohammad Abdullah Al Abdullah, Khalid Mohammed Alzahrani, Ammar Abdulmohsen Allehyani.

**Writing – review & editing:** Khalid Talal Aboalshamat, Mohammed Osamah AlHajeeri, Mohammad Abdullah Al Abdullah, Khalid Mohammed Alzahrani, Ammar Abdulmohsen Allehyani.

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
