## [Decision Letter · Decision Letter 0]

27 Aug 2025

PONE-D-25-21706What Gets a Dentist Hired? Factors influencing employment of recent dentist graduates in Saudi ArabiaPLOS ONE

Dear Dr. Aboalshamat,

Thank you for submitting your manuscript to PLOS ONE. After careful consideration, we feel that it has merit but does not fully meet PLOS ONE’s publication criteria as it currently stands. Therefore, we invite you to submit a revised version of the manuscript that addresses the points raised during the review process.

We look forward to receiving your revised manuscript.

Kind regards,

Ayesha Fahim

Academic Editor

PLOS ONE

Journal Requirements:

3. We notice that your supplementary information is uploaded with the file type 'Other”. Please amend the file type to 'Supporting Information'. Please ensure that each Supporting Information file has a legend listed in the manuscript after the references list.

4. Please remove all personal information, ensure that the data shared are in accordance with participant consent, and re-upload a fully anonymized data set.

Reviewers' comments:

Reviewer's Responses to Questions

**Comments to the Author**

1. Is the manuscript technically sound, and do the data support the conclusions?

Reviewer #1: Yes

Reviewer #2: Partly

2. Has the statistical analysis been performed appropriately and rigorously? 

Reviewer #1: Yes

Reviewer #2: Yes

3. Have the authors made all data underlying the findings in their manuscript fully available?

Reviewer #1: Yes

Reviewer #2: Yes

4. Is the manuscript presented in an intelligible fashion and written in standard English?

Reviewer #1: Yes

Reviewer #2: Yes

5. Review Comments to the Author

Reviewer #1: Report – Manuscript PONE-D-25-21706

Dear Editor,

Thank you for the opportunity to review the manuscript titled “What Gets a Dentist Hired? Factors influencing employment of recent dentist graduates in Saudi Arabia”. The study addresses a relevant and timely issue, particularly in the context of rising numbers of dental graduates and employment challenges in the health sector. While the authors attempt to contribute meaningfully to the literature by identifying employment correlates among Saudi dental graduates, there are notable methodological and interpretive issues that require attention before the manuscript can be considered for publication.

Overall Assessment

The study employs a cross-sectional design using a self-developed questionnaire distributed via social media to recent dental graduates. While the topic is important, several concerns limit the generalizability and interpretive power of the findings. These concerns relate primarily to sampling design, insufficient methodological clarity, and overreaching in data interpretation.

Methodological Concerns and Clarity

The authors justify their use of convenience sampling due to a lack of a sampling frame, but do not adequately explain the rationale behind the four-month post-graduation inclusion criterion. Further, the recruitment approach, described as distribution via social media, is vague. It remains unclear whether participants were contacted through individual outreach, open group posts, or targeted advertising. This has implications for understanding who was likely to respond and how representative the sample is of all recent graduates.

The description of questionnaire development is similarly ambiguous. The manuscript does not clearly identify which items were adapted from existing studies and which were developed by the authors. References to a group of five dental professionals involved in item generation lack context, such as their expertise or affiliations. Terms like “classified according to the results” also need clarification. Additionally, although the authors claim content validity was established through a pilot with 17 professionals, more detail on the outcome of that process is needed.

Questionable Inferences and Over-Interpretation

The manuscript makes several claims that extend beyond what the sampling and analysis can support. For example, it states that individuals with higher GPAs were “more likely” to be employed in government positions. Given the non-probabilistic sampling, such language implies causality or population-level generalizability that is not warranted. In other instances, findings like the association between marital status and employment are interpreted in ways that are culturally and temporally ambiguous (e.g., suggesting marriage may lead to employment or vice versa), without acknowledging reverse causation or third variables.

Similarly, findings from bivariate tests are discussed as definitive patterns rather than preliminary associations, and comparisons to earlier studies (e.g., Farag et al.) are made without considering differences in design, sample population, or recruitment approach.

Measurement and Validity Issues

Several variables—particularly those related to soft skills, job applications, and social media activity—are based on self-report without any control for social desirability or independent validation. This is especially important given the nature of questions such as “Would you do a procedure you’re not licensed to perform?” The ethical implications of such items are concerning, and the framing of these questions warrants reconsideration. The authors should also reflect on how reliable these self-assessments are and how they affect conclusions drawn about employability.

Contradictory or Misleading Interpretations

The manuscript includes internal contradictions. For example, it asserts that social media activity does not influence employability, yet cites other studies showing its importance in patient acquisition. These are not contradictory findings per se, since they refer to different outcomes (employment vs. client-building). However, the manuscript presents them as opposing, weakening the argument. Likewise, the phrase “more likely to be employed” is used repeatedly despite the convenience sampling and lack of controls for confounding variables. At minimum, this language should be revised to indicate associations rather than predictive relationships.

Recommendations and Limitations

The authors acknowledge several limitations at the end of the paper. However, many of the inferences and policy recommendations throughout the manuscript do not reflect these limitations. If the authors are to advocate for policy changes, such as adjustments to hiring practices or salary structures, those claims should be based on more rigorous and representative data, or at least tempered in tone.

While the manuscript brings important attention to a growing workforce issue in Saudi Arabia, its current form requires major revision. The authors should clarify their methodology, temper their conclusions, and improve the transparency of their questionnaire design. Future revisions should also avoid overstatement of findings and better align interpretation with methodological constraints.

I appreciate the opportunity to provide feedback on this manuscript and hope the comments will help the authors in strengthening their work.

Thank you.

Reviewer #2: Keywords must be based on MESH TERM.

Regarding sample size, reference should be made to a specific article.

There is no report on the validity and reliability of the questionnaire.

In the discussion section, the beginning should be different and it is not necessary to write exactly the conclusion contents.

The discussion needs more attention and revision, and relevant articles should be used, not necessarily related to the country of Saudi Arabia.

References should be rechecked and matched to the journal format.

Questionnaire should be added.

6. PLOS authors have the option to publish the peer review history of their article (what does this mean? ). If published, this will include your full peer review and any attached files.

**Do you want your identity to be public for this peer review?** For information about this choice, including consent withdrawal, please see our Privacy Policy .

Reviewer #1: No

Reviewer #2: No

---

## [Author Response · Author response to Decision Letter 1]

28 Aug 2025

Dear editor,

We have attached a file containing responses to the reviewers' comments. Thank you.

---

## [Editor Report · Decision Letter 1]

14 Sep 2025

What Gets a Dentist Hired? Factors influencing employment of recent dentist graduates in Saudi Arabia

PONE-D-25-21706R1

Dear Dr. Aboalshamat,

We’re pleased to inform you that your manuscript has been judged scientifically suitable for publication and will be formally accepted for publication once it meets all outstanding technical requirements.

Kind regards,

Ayesha Fahim

Academic Editor

PLOS ONE
---

## [Editor Report · Acceptance letter]

PONE-D-25-21706R1

PLOS ONE

Dear Dr. Aboalshamat,

I'm pleased to inform you that your manuscript has been deemed suitable for publication in PLOS ONE. Congratulations! Your manuscript is now being handed over to our production team.

Kind regards,

on behalf of

Dr. Ayesha Fahim

Academic Editor

PLOS ONE